# Lignin-Based Phenolic Foam Reinforced by Poplar Fiber and Isocyanate-Terminated Polyurethane Prepolymer

**DOI:** 10.3390/polym13071068

**Published:** 2021-03-28

**Authors:** Guoliang Chen, Jian Liu, Wei Zhang, Yanming Han, Derong Zhang, Jianzhang Li, Shifeng Zhang

**Affiliations:** 1Key Laboratory of Wood-Based Materials Science and Utilization, Beijing Forestry University, No. 35 Tsinghua East Road, Haidian District, Beijing 100083, China; guoliangchen_bjfu@163.com (G.C.); colinchen@bjfu.edu.cn (J.L.); zhdr666666@sina.com.cn (D.Z.); lijzh@bjfu.edu.cn (J.L.); hnzsf142@126.com (S.Z.); 2Beijing Key Laboratory of Wood Science and Engineering, Beijing Forestry University, No. 35 Tsinghua East Road, Haidian District, Beijing 100083, China; 3Research Institute of Forestry New Technology, Chinese Academy of Forestry, Xiangshan Road, Beijing 100091, China

**Keywords:** polyurethane prepolymer, hybrid reinforcement, phenolic foam, anti-pulverization

## Abstract

Phenolic foams (PFs) are lightweight (<200 kg/m^3^), high-quality, and inexpensive thermal insulation materials whose heat and fire resistance are much better than those of foam plastics such as polyurethane and polystyrene. They are especially suitable for use as insulation in chemical, petroleum, construction, and other fields that are prone to fires. However, PFs have poor mechanical properties, poor abrasion resistance, and easy pulverization. In this paper, a polyurethane prepolymer was treated with an isocyanate, and then the isocyanate-terminated polyurethane prepolymer and poplar powder were used to prepare modified lignin-based phenolic foams (PUPFs), which improved the abrasion resistance and decreased the pulverization of the foam. The foam composites were comprehensively evaluated by characterizing their chemical structures, surface morphologies, mechanical properties, thermal conductivities, and flame-retardant properties. The pulverization ratio was reduced by 43.5%, and the thermal insulation performance and flame-retardancy (LOI) were improved. Compared with other methods to obtain lignin-based phenolic foam composites with anti-pulverization and flame-retardant properties, the hybrid reinforcement of foam composites with an isocyanate-terminated polyurethane prepolymer and poplar powder offers a novel strategy for an environmentally friendly alternative to the use of woody fibers.

## 1. Introduction

Phenolic foams (PFs) are widely used in the exterior wall insulation of buildings, thermal pipeline transportation in chemical factories, and liquid natural-gas ship insulation due to their numerous advantages, including good flame-retardant properties and low smoke development [1,2]. Phenolic molecules only contain carbon, hydrogen, and oxygen atoms; thus, when subjected to high temperatures, they decompose and form a small amount of CO gas, but no other toxic gases. The maximum smoke density is 5.0%, and after a 25 mm-thick phenolic foam board was exposed to a flame spray of 1500 °C for 10 min, its surface was only slightly carbonized without burning or emitting dense smoke or poisonous gases. Moreover, PFs have good performance and thermal stability between 196 and 200 °C. They also have a low thermal conductivity (0.023 W/m·K) and are, therefore, widely used as insulation [3]. They also have strong resistance to chemicals and solvents.

Despite these advantages, PFs exhibit relatively poor mechanical properties (compressive strength is 119 kPa; compression modulus is 3674 kPa) and undergo extreme abrasion (pulverization ratio is 9.2%) compared with other polymer foams [4]. In recent years, to overcome the deficiencies of traditional PFs and decrease environmental pollution, many researchers have investigated the use of natural fillers and reinforcements (such as wood flour, pulp fiber, flax, bagasse, and lignin) [5,6,7,8]. In addition, PFs also have shortcomings in terms of raw materials and their own characteristics, which restrict their large-scale applications. Due to their expensive petroleum-based raw materials, such as phenol, their cost is relatively high [9,10,11,12]. Additionally, due to the structural characteristics of phenolic resins, there are many rigid methylene bridge bonds in pure phenolic foams, which make the resulting foam material brittle, which affects their construction applications. Faced with these problems, researchers have used biomass substitution methods to modify the molecular structure and multiple complexes of PF [13,14,15,16]. 

Many researchers have modified PFs with lignin to obtain foam materials with improved performance [17,18,19]. For example, Zhang et al. found that the introduction of lignin reduced and homogenized the pore size of a foam, which significantly improved its properties [20,21,22]. Lignin, tannin, etc., which are inexpensive and widely found in nature, can be used to partially replace phenol due to their similar molecular structures, thereby allowing for the preparation of biomass-based PFs to reduce costs. In addition, some researchers have studied the use of wood fiber and glass [23,24,25]. The use of fibers in PFs via multiphase compositing has been found to improve the anti-pulverization performance of biomass-based PFs. Zhang et al. introduced silicon whiskers into foams, and the uniform distribution of the silicon whiskers further improved the thermal stability and mechanical properties of the foam [26]. Song et al. used multi-walled carbon nanotubes (MWCNTs) to modify a lignin-based PF, which was found to have higher compressive and flexural strength than pure PF. MWCNTs can produce a micro-level dense carbon layer when the foam burns, which isolates external heat and air and prevents lignin from reducing the flame-retardant properties of the foam [27]. Early research included investigations by Li [28] and Liu [29], who studied the enhancement and toughening of lignin- and tannin-modified PFs and wood flour biomass fillers; however, these rigid materials tended to be more brittle. Fillers, such as wood flour and wood fiber, have insufficient toughness and poor two-phase interface compatibility; therefore, a fundamental solution to this problem is to modify the molecular structure and introduce flexible molecular segments to toughen their structure.

Many researchers have studied the molecular toughening modification of phenolic resins and foams. Ding et al. obtained a poly(ethyl carbamate) prepolymer by the Mannich reaction and esterification reaction. The PF modified by this quasi-prepolymer was found to have improved compressive and flexural strength, as well as a reduced apparent density [30]. Xu et al. prepared a polyurethane (PU) prepolymer with diphenyl-methane-diisocyanate(MDI) and then modified the PF with this prepolymer and H_3_BO_3_. When 8% PU prepolymer was added, the compressive and tensile strengths of the modified PF reached their maximum; however, if this PU amount was exceeded, the mechanical properties of the modified PF were better than those of pure PF [31]. In the research by Yang et al., a novel phosphorus- and silicon-containing polyurethane prepolymer was synthesized by the reaction of phenyl dichlorophosphate with hydroxyl-terminated polydimethylsiloxane and then with toluene-2,4-diisocyanate. After the PF was modified with the prepolymer, it maintained excellent fire resistance and exhibited increased compressive and impact strengths with a reduced comminuting rate [32]. Yang et al. used a resole phenolic resin, surfactants, blowing agents, PU prepolymers, short glass fibers, and other materials to prepare modified PFs, which exhibited improved anti-pulverization properties and reduced abrasion compared with the pure PF [33]. While the performance of PU-modified phenolic resins has been significantly improved, these modifications are accompanied by increased costs. 

In our previous study, we used poplar flour to improve the anti-pulverization properties of a lignin-based phenolic foam that contained a large number of hydroxymethyl groups and hydroxyl groups. A condensation reaction occurs between the hydroxymethyl functional group and the active site or hydroxymethyl on the benzene ring during foaming molding. Methylene bridge bonds and short-chain ethers are the main intermolecular bonds. When the foam was subjected to an external force, the activity of the molecular chains was relatively weak and inhibited, which limited the strain. Lignin-based phenolic foams contain a large number of rigid structures (benzene rings), which makes them brittle. Such foams rely only on external toughening, which does not solve the pulverization problem. The purpose of this work was to obtain a low-cost, environmentally friendly, biomass-based PF with excellent anti-pulverization properties. Here, a low-cost lignin phenolic resin was synthesized and then composited with low-cost poplar wood powder to create a physical network. Most importantly, a high-activity isocyanate-terminated PU prepolymer was synthesized with a molecular polymer and lignin phenolic resin, which were copolymerized to obtain a low-cost, environmentally friendly, biomass-based PF with a significant toughening effect.

## 2. Materials and Methods

### 2.1. Materials

Phenol, sodium hydroxide, concentrated sulfuric acid (98%), formaldehyde, paraformaldehyde, phosphoric acid, Tween-80, petroleum ether, *p*-toluenesulfonic acid, hydrochloric acid (37%), boric acid, and oxalic acid were purchased from Beijing Chemical Reagent Co., Ltd., and were of analytical grade.

Poplar flour was industrial grade (Beijing Forestry University Wood Factory, Beijing, China), with a particle size of less than 0.15 mm.

Industrial-grade diphenylmethane diisocyanate (MDI, -NCO mass fraction of 32.6%) and polyoxypropylene glycol (PPG; *M*_n_ = 2000) were purchased from Wanhua Chemical Group Co., Ltd., Yantai, China.

### 2.2. Preparation of Phenolated Depolymerized Lignin (DL)

Phenol (80 g) and sodium hydroxide were used as catalysts; 5 wt% (phenol + lignin) and 20 g lignin were added to four 500 mL flasks, which were quickly heated to 90–92 °C for 1, 2, 3, and 4 h and static to room temperature. The samples were washed repeatedly with ether until phenol could not be detected in concentrated bromine water. The washed samples were precipitated with dilute hydrochloric acid, then washed with distilled water until the pH was neutral. The precipitate (phenolic lignin) was then dried in a vacuum drying oven at 50 °C for 24 h.

### 2.3. Preparation of DL-Based PF (DLPF)

The reaction (Figure 1) was performed at 82 °C for 30 min, followed by the addition of the required amount of phenolic lignin, 103 g paraformaldehyde, 180 g formaldehyde solution, and 20 g NaOH (40 wt%) solution. Phenolated lignin-based phenolic resin with 20% phenolated lignin were obtained and denoted as DLPR-20 (DL-based phenolic resin-20).

At room temperature, 140 g resin was placed into a plastic beaker, to which 40 g (95%) phosphoric acid was also added. This beaker was stirred at 500 rpm for 5 min, and then 14 g surfactant (Tween-80) was added. Then, 20 g petroleum ether and 20 g composite acid curing agent (*p*-toluenesulfonic acid/phosphoric acid/hydrochloric acid/water = 2:1:2:1) were added. Each time a chemical was added, the mixture was stirred at 500 r/min for 5 min. The mixture was then poured into a foaming mold (2 L) with a lid and cured in an oven at 80 °C for 2 h. The resulting phenolated lignin-based phenolic resin foam was denoted as DLPF-20 according to the lignin replacement rate.

### 2.4. Preparation of Blocked Isocyanate-Terminated PU Prepolymer

Polyether polyol was dehydrated at 110 °C and a vacuum pressure of 0.1 MPa. When the moisture content of PPG was less than 0.005%, dehydration was stopped. The dehydrated PPG was slowly dripped into MDI at 55 °C. After dripping, the mixture was heated to 78 °C for 2 h, and the entire reaction was completed under a nitrogen atmosphere. The isocyanate-terminated PU polymer was obtained. According to the different molar ratios of MDI to PPG (1.5, 2.0, 3.0, and 4.0), the prepared PU prepolymers were marked respectively as PU-1.5, PU-2, PU-3, and PU-4, as shown in Table 1.

### 2.5. Characterization of Blocked Isocyanate-Terminated PU Prepolymer

The isocyanate content in the PU was determined by the acetone-di-*n*-butylamine method. The dried PU was dissolved in tetrahydrofuran (THF). The weight-average molecular weight (*M*_w_), number-average molecular weight (*M*_n_), and relative molecular weight distribution index (*M*_w_/*M*_n_) of PU were determined by gel permeation chromatography with THF as the mobile phase and a flow rate of 1 mL/min.

### 2.6. Preparation of DLPF-20 Modified by PU with Different Proportions and 1% Poplar Wood Powder

DLPF-20, 1% poplar wood powder, and the designated amount of PU (as shown in Table 2) were added to a 250 mL beaker. Stirring was then conducted at 600 rpm for 15 min to obtain a homogenous mixture.

At room temperature, an appropriate amount of resin was placed into a plastic beaker, to which phosphate (95%) was added. The mixture was stirred at 500 rpm for 5 min, after which a surfactant (Tween-80), petroleum ether, and a compound acid curing agent (*p*-toluene sulfonic acid/phosphoric acid/hydrochloric acid/water = 2:1:2:1) were added to the beaker. After each chemical was added, the mixture was stirred at 500 r/min for 5 min. The mixture was poured into the foaming mold (2 L) with a lid and solidified in an oven at 80 °C for 2 h.

The PU/poplar powder-reinforced lignin-based phenolic resin foams were respectively denoted as PUPF0, PUPF2, PUPF5, PUPF7, and PUPF9 according to the amount of PU.

### 2.7. Characterization of FDLPF 1 Modified with Different Proportions of PU

Fourier-transform infrared (FTIR) spectroscopy was used to characterize the foam composites using ATR spectroscopy.

The apparent density of the foam composites was measured according to the GB/T6343-2009 standard [34]. The sample size was 30 × 30 × 30 mm^3^. Care was taken during cutting to not destroy the original cell structure.

Scanning electron microscopy (SEM) was performed with a Hitachi S-4800 to observe the foam composites.

According to the ASTMD(American Society for Testing and Materials) D1621-10 standard, measurements were carried out at room temperature by a universal testing machine (Instron6022, the maximum capacity of the load-cell was 50 kN) [35]. The specimen size was 30 × 30 × 30 mm^3^, and the compression velocity was 2.5 mm/min. When the strain of the specimen was less than 10%, the compressive strength and modulus of each specimen were recorded. Measurements were repeated five times for each sample, and the average was used as the final value. The compressive strength was calculated by Formula (1).
(1)$$\sigma_{m}=10^{3}\cdot \frac{F_{m}}{A_{0}}$$

*F*_m_—The maximum compressive force when the relative deformation is less than 10% (N)

*A*_0_—The initial cross-sectional area of the sample (mm^3^)

Each foam composite was cut into samples with dimensions of 50 × 50 × 20 mm^3^ and weighed (*M*_1_). The edges of each sample were placed on sandpaper, and 200 g weights were placed on the foam composite. The samples were pulled back and forth across the sandpaper 30 times, each for a distance of 250 mm [36]. The mass of the sample was then weighed (*M*_2_). The crushing rate was calculated as (*M*_1_–*M*_2_)/*M*_1_×100%. Four samples of each foam composite were tested and averaged.

The limiting oxygen index (LOI) of each sample was determined by an HC-2 LOI meter according to the ASTMD D2863 standard [37]. The foamed plastics were cut into 100 × 10 × 10 mm^3^ samples, and the reported LOI of each sample was the average value of five samples.

The thermal conductivity of the foam composites was measured with a Hot Disk TPS 2500 thermal-conductivity analyzer. The size of each sample was 30 × 30 × 15 mm^3^. Three samples of each foam composite were tested, and the average was reported.

## 3. Results

### 3.1. Optimization of PU Prepolymer

After testing(Table 3), it was found that the isocyanate concentration and molecular weight of the PU prepolymer increased upon increasing the amount of MDI. When the amount of MDI was 2 mol, the molecular weight distribution of the PU-2 prepolymer was moderate, and the distribution range was the narrowest, making it suitable for the subsequent modification and preparation of foam materials. The preparation of PU prepolmers is shown in Figure 2.

The acetone-di-*n*-butylamine method was used to determine the isocyanate content in PU. A small amount of polyurethane (accurate to 0.0001 g) was dissolved in 10 mL of acetone. After it was completely dissolved, 10 mL of acetone-di-*n*-butylamine solution was added. After shaking, three drops of bromocresol green were added as an indicator. After standing for 15 min, the solution was titrated with standard 0.1 M hydrochloric acid. The indicator changed from green to yellow as hydrochloric acid was consumed. This procedure was repeated three times for each group. The -NCO content was calculated using Formula (2).
(2)$$\mathrm{NCO}\%=\frac{\left(V1-V0\right)\cdot C\cdot 42.02\cdot 100}{1000\cdot m}$$

*V*_0_—Blank consumption HCl volume (mL)

*V*_1_—Sample consumption volume of HCl (mL)

*C*—HCl concentration (mol/L)

*m*—Sample weight (g)

42.02 g/mol—Molar mass of NCO group

100—Convert the calculation result to “%”

1000—Convert “g” to “mg”

### 3.2. Mechanism Analysis of PU Prepolymer/Poplar Wood Flour Hybrid Enhanced PF

Some studies have shown that the active groups in PU prepolymers can be crosslinked with the hydroxymethyl groups of phenolic resins; thus, flexible PU groups are introduced into the rigid phenolic resin to toughen the phenolic foam. A PU prepolymer that does not react with the phenolic resin can also penetrate the network of the phenolic resins, and the interpenetration and entanglements between PU chains and the phenolic resin increased the viscosity of the system and interfacial interactions, thus improving the anti-pulverization of the phenolic foam [38,39,40]. To toughen the lignin-based PF, PU was used as an internal toughening material, and poplar wood powder was used as an external toughening material. The toughening mechanism of the PU/poplar wood flour is summarized as follows. PU participated in the foaming process via chemical grafting, and residual isocyanate radicals and hydroxyl groups in the resin base material remained after this process. Hydroxymethyl groups reacted by grafting long PU chains into the PF. The long PU chains reduced the density of crosslinking points in the crosslinked phenolic resin. After poplar wood flour was added, its own strength could support the pore wall structure of the foam. The nucleation of wood flour helped form a uniform cell structure. The wood powder and resin, which were closely crosslinked, worked together when exposed to an external force to disperse the stress, which improved the performance of the material. The mechanism of PU/poplar wood flour hybrid enhanced phenolic foam (PF) is shown in Figure 3.

Figure 4 presents the FTIR spectra that show the effects of adding PU to the lignin-based foams. After adding PU, the toughened PF did not exhibit the absorption peak for an isocyanate group C=N at about 2260 cm^−1^. Instead, the characteristic absorption peak of C=O in urethane appeared at 1619 cm^−1^. After adding more PU, the peak gradually became stronger, which indicates that the number of carbamate C=O bonds in the foam increased. The chemical peak changes prove that the isocyanate groups in PU reacted with hydroxy and hydroxymethyl groups in lignin-based phenolic resins to form carbamates, in addition to their reactions with functional groups on wood fibers.

### 3.3. Morphology of Hybrid Enhanced PF

The characteristics of the modified lignin-based foam after adding PU are shown in Table 4. It was found that upon increasing the amount of PU, PUPF2, PUPF7, and PUPF9 exhibited slight increases in their apparent density compared with PUPF0, while PUPF5 remained unchanged.

Figure 5 shows that the cell diameter of the PU-modified foams was smaller, and the cells were tightly connected. When 5% PU was added, the median cell diameter reached a minimum of 93.97 μm, and the cell density reached a maximum of 6.24 × 10^5^ cells/cm^3^. At this time, the cell structure was relatively uniform, and the compressive strength reached a maximum of 125.125 kPa.

Phenolic foams are a kind of fibrous porous medium. It has been previously shown that fibrous porous media can be described as fractal porous media, and some properties of the porous media are affected by their microstructural parameters [41].

### 3.4. Mechanical Properties of Hybrid Enhanced PF

When a PF is under pressure, the key factor that affects its compressive strength is whether the stress acting on the internal structure of the foam can be effectively dispersed. Figure 6 shows that the compressive strength and modulus of the lignin-based PF first increased and then decreased upon increasing the PU content. The compressive strength (125.13 kPa) and modulus (3313 kPa) of the lignin-based PF modified with 5% PU were the maximum, which indicates excellent compression and anti-pulverization properties. Since the cell size of PUPF5 is smaller than other PUPFs, the stress between the cells will be better dispersed when subjected to external forces, which improves the mechanical properties [42,43,44,45]. 

Figure 7 shows the compressive stress–strain curves of the PF and PUPFs. Similar to plastic foams, PUPFs exhibited a multi-stage deformation response when subjected to compressive loading. The stress–strain curves of the samples can be divided into three regions: elastic region, yield region, and compaction region. At the initial loading stage, the strain increased linearly upon increasing the stress (elastic region). The elastic deformation of the foam wall, such as bending of the edges of the foam wall, compression of gases in closed pores, and stretching of the foam wall, mainly occurred, reflecting the elastic properties of the material and the foam strength. Upon further increasing the load, most of the foam began to lose its stability, entering the yield zone, in which weak areas of the wall, such as fiber perforations, broke. In addition, due to debonding of the interface between fiber and resin, cracks appeared and continued to expand in the foam, which reflects the crushing of the material. The length of the yield zone and the stress level can be used to evaluate the material’s cushioning and energy-absorbing properties. As the foam and reinforcement fibers were crushed further, the material entered the compaction zone, in which the compressive stress increased abruptly upon increasing the strain. When more than 5% PU was added, the compressive strength and modulus of the PU-modified PF tended to decrease to varying degrees; however, when 9% PU was added, the compressive strength and modulus decreased greatly. The reason for the increased energy is the large amount of PU added, which decreased the foam wall strength. When the amount of added prepolymer was low, the prepolymer produced good crosslinking with the foam structure, introduced flexible segments, and improved its performance. When too much prepolymer was added, the foaming was affected, and the cell structure was too large. In addition, because the curing conditions of the polyurethane were different from those of the phenolic aldehyde, excess additive negatively impacted the foam system.

Figure 8 presents the pulverization ratios of the PF, PUPF0, PUPF2, PUPF5, PUPF7, and PUPF9. The pulverization ratios first decreased and then increased upon increasing the polyurethane content. When the polyurethane content was 5%, the pulverization ratios of PUPFs reached a minimum, which may be due to the grafting of long-chain polyurethanes onto the lignin phenolic foam, which extended the average distance between active sites. When subjected to an external force, the polyurethane-modified foams showed greater strain, which helped reduce the fracture of the material. At the same time, the addition of the PU prepolymer reduced the proportion of benzene rings in the material’s structure, which reduced the powdering rate of the foam; however, when excess polyurethane was added, the strength of the foam wall decreased, which caused the overall performance of the material to decline.

### 3.5. Thermal Properties of Hybrid Enhanced PF

The thermal conductivities of modified PFs with different amounts of PU are presented in Figure 9. The thermal conductivities of PUPF2, PUPF5, PUPF7, and PUPF9 were 0.049, 0.048, 0.050, and 0.053 W/(m·K), respectively. It has been pointed out that the thermal conductivity of porous media with rough surfaces is related to their microstructural parameters. The thermal conductivity of porous media with rough surfaces decreases upon increasing the relative roughness and zigzag fractal dimension [46]. Compared with the foam without PU (PUPF0), there was no significant change in the thermal conductivity, which indicates that the addition of PU did not have a large impact on the cell structure.

Figure 10 presents a histogram of the LOIs of PFs modified with different amounts of PU. The LOIs of PUPF2, PUPF5, PUPF7, and PUPF9 were 33.7%, 32.5%, 30.2%, and 28.7%, respectively. The LOI of the PU-modified lignin-based PFs decreased upon increasing the PU amount, which was mainly caused by the inflammability of PU; however, when 5% PU was added, the flame-retardant properties of the lignin-based PFs remained excellent, and the PF could be characterized as a B1 flame-retardant material.

According to ASTM E-1354, the combustion performance of the samples was tested at a radiation power of 50 kW/m^2^ using 100 mm × 100 mm × 5 mm samples. The test parameters included ignition time (*T*_TI_), peak heat release rate (*P*_HRR_), total heat release (*T*_HR_), average specific extinction area (*A*_SEA_), and average mass loss rate (*A*_MLR_). Table 5 shows that adding the PU prepolymer decreased the flame-retardant properties of the foam, which is consistent with the LOI test results. If the ignition time of the material is longer, the peak heat release rate decreases, and the total heat release decreases, then the maximum energy release value of the material during combustion decreases, and the flame-retardant performance is better. As shown in Figure 11, to more intuitively show the adverse effect of the PU prepolymer on the flame-retardant properties of PUPF0, we investigated PUPF5, which showed the best performance with regards to the properties mentioned above, to compare its heat release rate curve with PUPF0. It can be seen that future research must determine how to overcome the negative impact of the flame-retardant properties of the polyurethane prepolymer while maintaining its strong anti-pulverization properties.

## 4. Conclusions

In this paper, the effects of adding different amounts of polyurethane to a phenolic foam were investigated. The mechanical properties, thermal insulation, and flame retardancy of the foam were characterized, and the mechanism by which the PU reinforced the phenolic resin foam was summarized.

PU prepolymer, a long-chain flexible segment, was chemically grafted onto a lignin-based PF, and it was found that the foam modified with 5% PU exhibited good performance. The strength of the foam modified with 5% PU increased from 119.6 to 125.13 kPa, its pulverization rate was reduced from 7.7% to 5.2%, and its LOI was reduced from 36.2% to 32.5% due to the properties of PU.

The main reasons for the toughening of the PFs modified with PU/wood powder were determined to be as follows. PU was included in the resin foam via chemical grafting. During grafting, the residual isocyanate groups in the PU reacted with the hydroxyl groups and methylol groups in the lignin-based phenolic resin. The long-chain PU reduced the density of crosslinking points in the phenolic resin network. Extending the distance between the reactive sites prevented the rigid fracture of the material under stress, and the addition of PU also reduced the proportion of rigid structures, such as benzene rings, in the material. Additionally, the strength of poplar wood flour supported the pore wall structure of the foam. The nucleation of wood flour helped the foam form a uniform cell structure. Wood flour and resin worked together when exposed to an external force to disperse stress, which improved the performance of the material.

Due to its poor anti-pulverization performance, the phenolic foam cannot be widely used as a building insulation material or flame-retardant due to the limitations of installation technology. Thus, in terms of production, the industrialization of phenolic foams has been limited. We hope that the research results of this article can be used to produce phenolic foams that are compatible with existing technology in the fields of conventional construction and wood structure construction and expand their applicability. We also hope that this article can provide insight and reference to researchers in related fields.

## Figures and Tables

**Figure 1 polymers-13-01068-f001:**
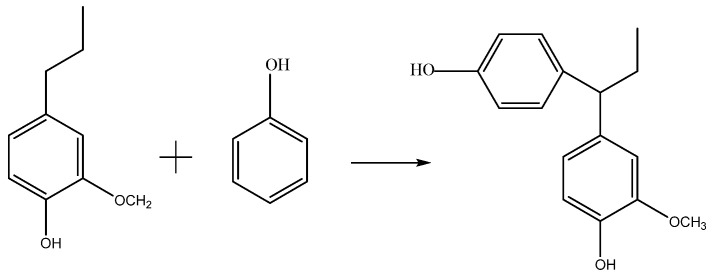
Reaction of lignin phenolation.

**Figure 2 polymers-13-01068-f002:**
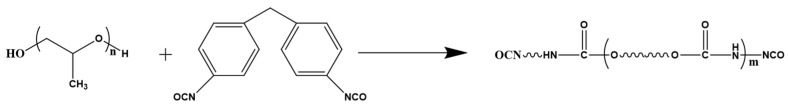
Preparation of PU prepolymers.

**Figure 3 polymers-13-01068-f003:**
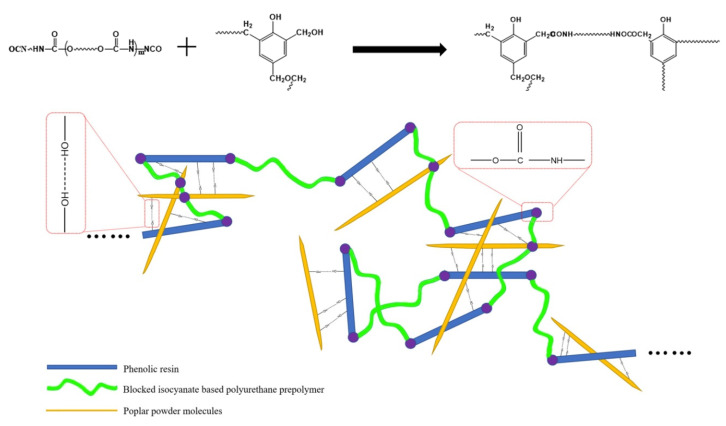
The mechanism of PU/poplar wood flour hybrid enhanced phenolic foam (PF).

**Figure 4 polymers-13-01068-f004:**
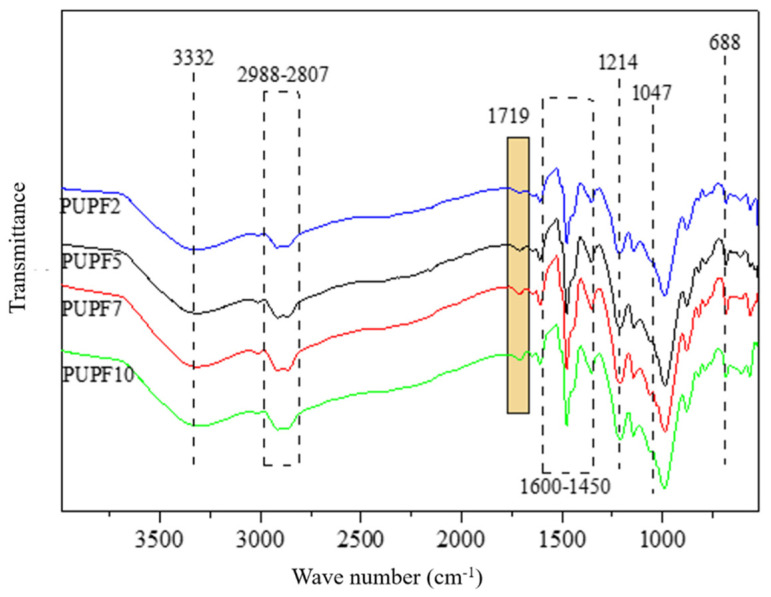
FTIR spectra of PU/poplar wood flour hybrid enhanced PF.

**Figure 5 polymers-13-01068-f005:**
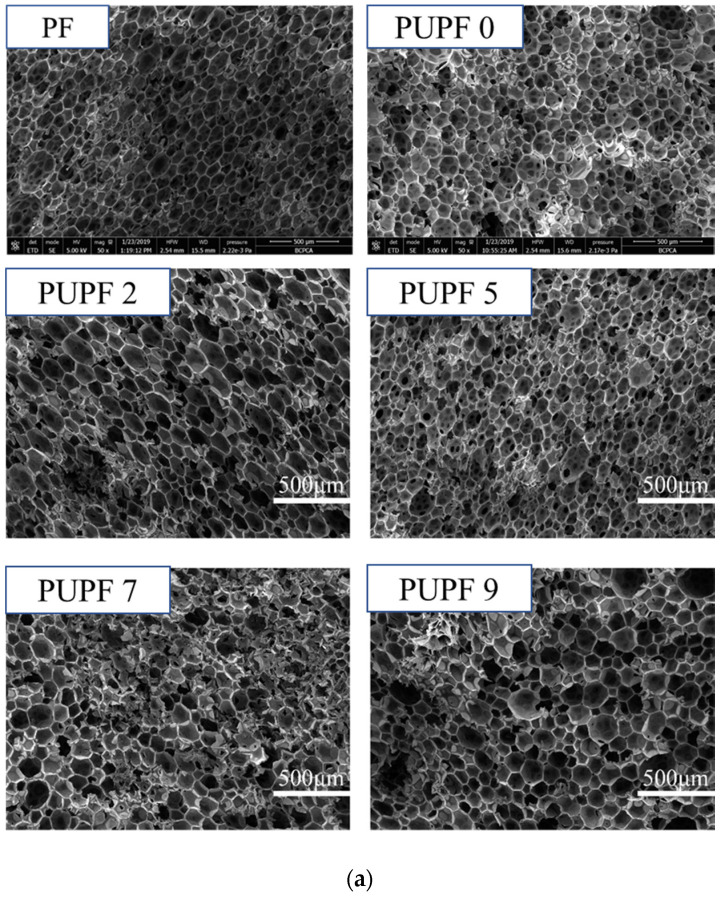
SEM images (**a**) of PF, PUPFs, and cell size distributions (**b**) of the PUPF2, PUPF5, PUPF7, and PUPF9.

**Figure 6 polymers-13-01068-f006:**
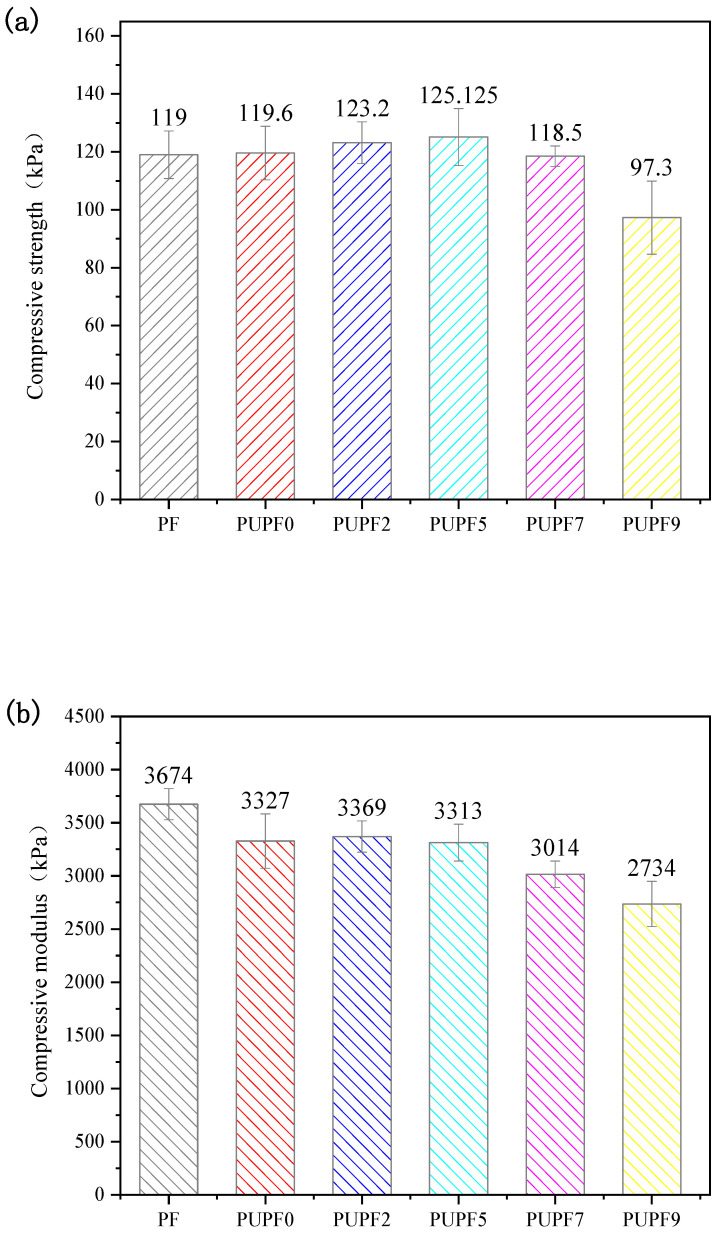
(**a**) Compressive strength and (**b**) compressive modulus of PF and PUPFs.

**Figure 7 polymers-13-01068-f007:**
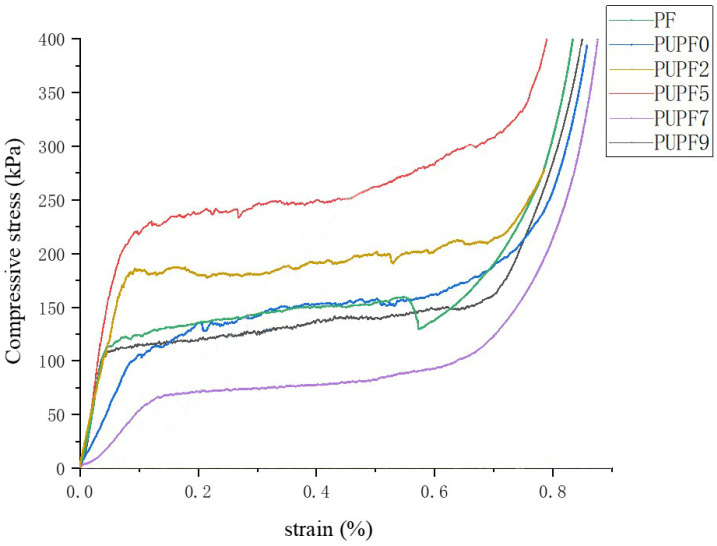
Compressive stress–strain curves.

**Figure 8 polymers-13-01068-f008:**
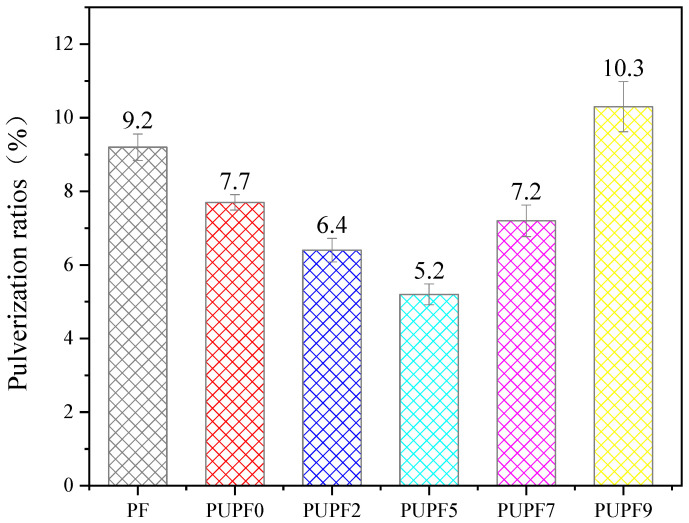
Pulverization ratios of PF and PUPFs.

**Figure 9 polymers-13-01068-f009:**
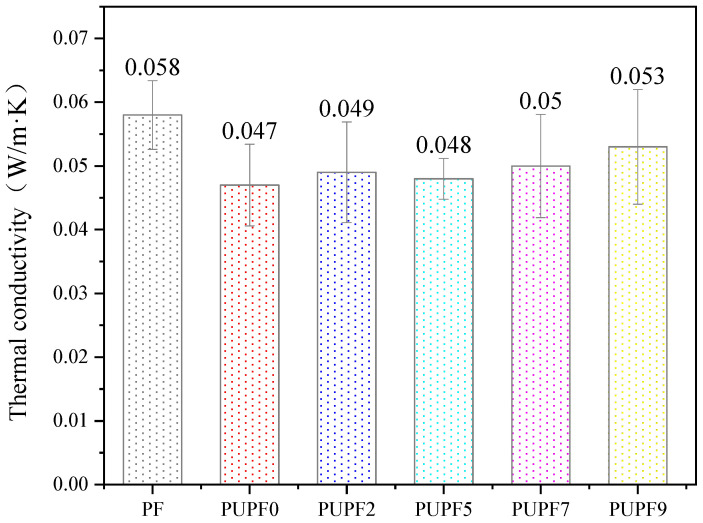
Thermal conductivity of PF and PUPFs.

**Figure 10 polymers-13-01068-f010:**
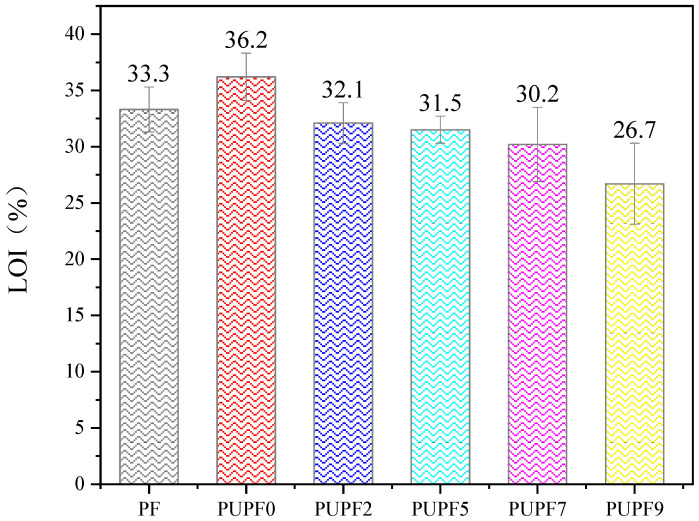
LOI of PF and PUPFs.

**Figure 11 polymers-13-01068-f011:**
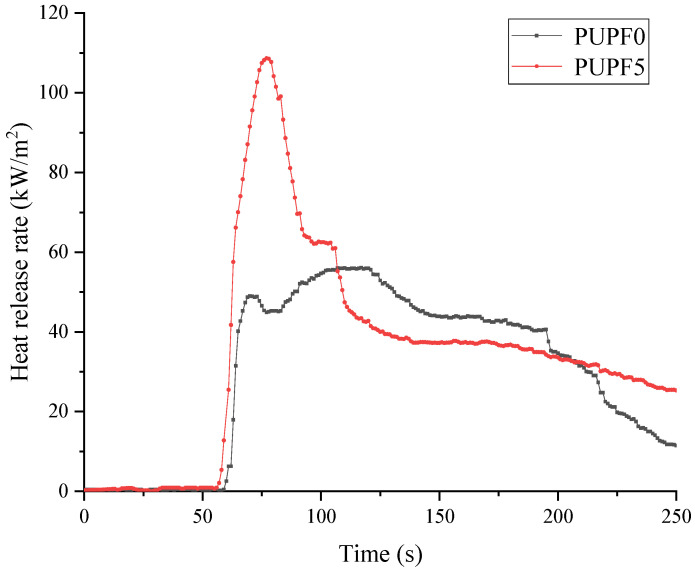
Heat release rate curves for PUPF0 and PUPF5.

**Table 1 polymers-13-01068-t001:** Preparation of blocked isocyanate-terminated PU prepolymers.

PU Prepolymer	PPG (mol)	MDI (mol)
PU-1.5	1	1.5
PU-2	1	2
PU-3	1	3
PU-4	1	4

**Table 2 polymers-13-01068-t002:** Formulation of foam samples.

Foam Name	Phosphoric Acid (%)	Tween-80 (%)	Vesicant (%)	Compound Acid Curing Agent (%)	Fiber (%)	PU-2 Content (%)
PF	13.3	13.3	6.67	13.3	0	0
PUPF0	13.3	13.3	6.67	13.3	1	0
PUPF2	13.3	13.3	6.67	13.3	1	2
PUPF5	13.3	13.3	6.67	13.3	1	5
PUPF7	13.3	13.3	6.67	13.3	1	7
PUPF9	13.3	13.3	6.67	13.3	1	9

**Table 3 polymers-13-01068-t003:** Characteristics of PU.

Sample	MDI (mol)	PPG (mol)	-NCO %
PU-1.5	1.5	1	1.8
PU-2	2	1	3.9
PU-3	3	1	5.7
PU-4	4	1	8.6

**Table 4 polymers-13-01068-t004:** Characteristics of PUPFs.

Sample Name	Apparent Density (kg/m^3^)	Median Cell Diameter (μm)	Cell Density *N_F_* (10^5^ cells/cm^3^)	Porosity (%)
PF	46.31 ± 1.19	103.31 ± 1.33	4.47 ± 0.23	94.27 ± 1.03
PUPF0	42.16 ± 2.18	93.97 ± 1.97	6.24 ± 0.14	93.07 ± 1.01
PUPF2	43.27 ± 1.73	94.36 ± 1.51	5.69 ± 0.21	93.79 ± 0.99
PUPF5	42.16 ± 2.21	93.97 ± 1.88	6.24 ± 0.31	93.07 ± 1.04
PUPF7	44.17 ± 2.36	97.32 ± 1.35	6.13 ± 0.19	92.13 ± 1.01
PUPF9	42.91 ± 1.93	96.31 ± 1.62	5.31 ± 0.11	91.87 ± 1.05

**Table 5 polymers-13-01068-t005:** Cone calorimetric data for PUPF0 and PUPF5 at a heat flux of 50 kW/ m^2^.

Sample	*T*_TI_ (s)	*P*_HRR_ (kW/m^2^)	*T*_HR_ (MJ/m^2^)	*A*_SEA_ (m^2^/kg)	*A*_MLR_ (g/s·m^2^)
PUPF0	4.3	55.98	7.64	229.1	2.166
PUPF5	3.6	107.32	9.10	114.67	2.048

## Data Availability

This study did not report any data.

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
