# Peer review of "Lignin-Based Phenolic Foam Reinforced by Poplar Fiber and Isocyanate-Terminated Polyurethane Prepolymer"

_polymers, 2021, doi:10.3390/polym13071068_

Round 1
Reviewer 1 Report
The paper is written well. I want to accept it after minor revision.
Following are some of the suggestions.
- Please add the reaction schematic for the reaction mentioned in the section 2.3.
- The authors have used PU as a toughening agent. Please add some proof saying that PU acts as toughnening agent.
Author Response
We thank you for the critical feedback. These valuable comments and suggestions helped us improve our manuscript. Based on the comments we received, the initial manuscript has been carefully modified. Changes in the revised version are marked in red text. We hope the new manuscript will meet your journal’s standards. You will find our point-by-point responses to the reviewers’ comments/ questions in the reply file.

Reviewer 2 Report
This work can be considered for publication in Polymers
Author Response
Thank you very much for your approval of this article.
Reviewer 3 Report
Phenolic foam is being applied worldwide to a variety of infrastructures owing to their excellent structural performance, durability, fire resistance, and economic efficiency. Besides, phenolic foam is widely used in the exterior wall insulation of buildings, thermal pipeline transportation in chemical factories, and liquid natural gas (LNG) ship insulation due to its numerous advantages. Its heat and fire resistance are much better than those of foam plastics, such as polyurethane and polystyrene. It is especially suitable for use as insulation in chemical, petroleum, construction, and other fields that are prone to experience fires. However, the greatest shortcomings of PF are its low mechanical properties, easy abrasion, and easy pulverization, which limit its wide use. In this manuscript, polyurethane prepolymer was treated with isocyanate, then the isocyanate-terminated polyurethane prepolymer and poplar powder was used to prepare modified lignin-based phenolic foams(PUPFs); this solved the phenomenon of easy abrasion and pulverization of the foam structure. The effects of different addition amounts of PU on foam were investigated. The mechanical properties, thermal insulation, and flame retardancy of the foam were characterized, and the mechanism of PU-reinforced phenolic resin foam was summarized. I am pleased to send you moderate comments. The results and theme of this paper is quite interesting. The layout is clear and easy to understand. Generally, this manuscript makes fair impression and my recommendation is that it merits publication in this Journal, after the following major revision:
- The authors need to reorganize the current introduction, which normally consists of three parts at least: background, literature review, brief of the proposed work. The current one is nothing but a literature review. Why their work is important comparing to previous reports? I think this is essential to keep the interest of the reader.
- Materials and Methods part. Although the results look “making sense”, the current form reads like a simple lab report. The authors should dig deeper in the results by presenting some in-depth discussion.
- The pulverization ratio was reduced by 43.5%, and the thermal insulation performance and flame retardant performance (LOI) were improved. The authors should give some explanation on above results and data (43.5%).
- In Fig.6 and 10, the authors should give the explanations for the difference of data collected from different sources.
- It is well known that phenolic foam is a type of porous media which has been widely used in many fields of life. In this manuscript, the foam composites were comprehensively evaluated via the characterization of their chemical structures, surface morphologies (see [A fractal model for capillary flow through a single tortuous capillary with roughened surfaces in fibrous porous media, Fractals, 2021, 29(1):2150017]), thermal conductivities (see [Fractals, 2020, 28(2): 2050029]), mechanical properties, and flame retardant properties. Authors should introduce some related knowledge to readers.
- Please, expand the conclusions in relation to the specific goals and the future work.
Author Response

(The authors gave the same response as above.)
